# HPRNA: Predicting synergistic drug combinations for angina pectoris based on human pathway relationship network algorithm

**Mengyao Zhou**[1,2☯], **Mengfan Xu**[1,3☯], **Xiangling Zhang**[4], **Xiaochun Xing**[4], **Yang Li**[5], **Guanghui Wang**[6], **Guiying Yan**[1,2]*

**1** University of Chinese Academy of Sciences, Beijing, China, **2** Academy of Mathematics and Systems Science, Chinese Academy of Sciences, Beijing, China, **3** Key Laboratory for Mechanics in Fluid solid Coupling Systems, Institute of Mechanics, Chinese Academy of Sciences, Beijing, China, **4** Tianjin Forth Central Hospital, Tianjin, China, **5** Tianjin Medical University Cancer Institute and Hospital, Tianjin Medical University, Tianjin, China, **6** Shandong University, Jinan, China

☯ These authors contributed equally to this work.

* yangy@amt.ac.cn

**Data Availability Statement:** All relevant data are within the manuscript and its Supporting information files.

## Abstract

Over the years, synergistic drug combinations therapies have attracted widespread attention due to its advantages of overcoming drug resistance, increasing treatment efficacy and decreasing toxicity. Compared to lengthy medical drugs experimental screening, mathematical models and algorithms show great potential in synergistic drug combinations prediction. In this paper, we introduce a novel mathematical algorithm, the Human Pathway Relationship Network Algorithm (HPRNA), which is designed to predict synergistic drug combinations for angina pectoris. We first reconstruct a novel angina pectoris drug dataset, which include drug name, drug metabolism, chemical formula, targets and pathways, then construct a comprehensive human pathway network based on the genetic similarity of the pathways which contain information about the targets. Finally, we introduce a novel indicator to calculate drug pair scores which measure the likelihood of forming synergistic drug combination. Experimental results on angina pectoris drug datasets convincingly demonstrate that the HPRNA makes efficient use of target and pathway information and is superior to previous algorithms.

## Introduction

### Background

The primary objective of modern drug research is to leverage the inhibitory impact of a drug on molecular targets in order to impede the entire disease progression, thereby accomplishing the goal of disease control or rehabilitation. However, with the development of known highly selective target ligands in recent years, the search for emerging drugs for these specific targets

**Funding:** This work was supported by the National Natural Science Foundation of China (Grant No. 12231018). The funder had no role in study design, data collection, data analysis, or preparation of the manuscript.

**Competing interests:** The authors have declared that no competing interests exist.

has become more and more difficult. New drug research and development faces complex processes, high economic costs, and challenges such as transient resistance [1], especially for complex diseases like cancer, cardiovascular and cerebrovascular diseases. For these reasons, synergistic drug combinations therapy has become the preferred method of disease treatment. In fact, synergistic drug combinations [2] refer to the combination of drugs can produce superposition effects greater than those after the single drugs are used separately. By targeting multiple pathways and molecular targets, these combinations can prevent the onset, transmission, and progression of disease related signals. Compared to single drugs, Synergistic drug combinations enhance efficacy and reduce toxicity by leveraging the complementary mechanisms of different medications, leading to better therapeutic outcomes and fewer adverse effects [3].

As we know, heart disease is a complex disease that seriously threatens human health, and it faces major challenges in treatment [4]. There are several types of heart disease, including arrhythmia and angina pectoris. Despite extensive research on angina, the pathogenesis of angina pectoris remains not fully understood, and the effectiveness of single drug therapies is often limited, so synergistic drug therapy is a natural and good choice. In conclusion, the study of combination drugs related to angina pectoris is of great significance for human health.

## Related work

The traditional drug combinations screening adopts the experimental method, they commonly used reference models such as the Loewe additivity model [5], isobologram model [6], and ChouTalalay model [7]. However, these traditional experimental methods are costly, time-consuming and poor efficiency. With the rapid advancement of computational technologies, various mathematical model-based methods for screening drug combinations have emerged. These computational approaches are becoming increasingly popular due to their ability to save time and reduce costs. They are particularly useful for predicting drug combinations, especially when information on potential combinations is limited, as in the case of new drugs. In the 2014 DREAM challenge on drug combinations, the DIGRE and IUPUI_CCBB models were among the best-performing methods, both relying on deterministic mathematical formulations. DIGRE effectively utilized a residual effect approach to model the complex relationships between genomic data and drug responses, providing accurate predictions based on mathematical principles. Similarly, the IUPUI_CCBB model employed a robust mathematical framework that integrated multiple biological layers, optimizing drug efficacy predictions through sophisticated mathematical modeling [8]. In recent years, a number of network-based approaches for drug combinations research have been proposed. Chen et al. [9] proposed a model named Network-based Laplacian regularized Least Square synergistic drug combination prediction (NLLSS), they predict potential synergistic antifungal drug combination and carried out biological experiment. Zou et al. [10] described the interactions between drug targets and their neighbors in the Protein-Protein interaction network, to distinguish synergistic drug combinations. They can not only predict the potential synergistic drug combinations, but also illustrate the latent synergistic molecular mechanism from the perspective of mutual interactions.

More recently, a growing number of drug combination screening methods based on machine learning and deep learning have emerged. For example, DeepSynergy [11] utilizes convolutional neural networks to predict the synergistic drug combinations. Hu et al. [12] combined transformer and graph neural networks for combination drug prediction. However, these methods require enough data, otherwise the effectiveness of these methods can be greatly

reduced. Additionally, a major limitation of these methods is their lack of interpretability, making it difficult to understand the underlying mechanisms driving the predictions.

Despite the success of these model-based approaches, most focus primarily on the targets of drugs as the core information. While they have demonstrated excellent performance in screening synergistic drug combinations for drugs with closely related targets, they fall short when it comes to drug combinations that act through broader biological networks. Wang et al. [13] observed that drug combinations are more likely to modulate function related pathways. Similarly, Zou et al. [10] found that drug association mainly acts on multiple targets of a pathway and its crosswalk path. Another study [1] on the molecular mechanisms of drug interactions suggests that pathway analysis is an effective approach for assessing drug combination effects. Based on this, we consider adding pathway analysis to reveal the underlying mechanism. The WWI method [14] used the information of pathway, but the network construction of this method is relatively complex so how to maintain the integrity of data utilization and reduce the complexity of network construction is the key to our research on synergistic drug combinations for angina pectoris.

So far, the synergistic drug combinations therapies for angina pectoris is common, mainly in the clinical practice of doctors. But we haven't found a quantitative mathematical way to predict the synergistic drug combinations for angina pectoris. Given that angina pectoris is a prevalent and intricate condition, posing a significant threat to human health, the need for research on drug combinations targeting angina pectoris cannot be overstated.

### Contributions

The contributions of this study can be summarized as follows: Firstly, we reconstructed a novel drug dataset related to angina pectoris by integrating data from multiple sources through database retrieval, which collect the names, chemical formulas, targets, and pathways of drugs used in the treatment of angina pectoris. Secondly, we proposed HPRNA to measure the correlation between two drugs, which includes the construction of human pathway relationship network and the calculation of the drug pairs' score. Finally, we apply the HPRNA to the dataset built above, which show excellent results compared with the previous algorithm.

### Notations and preliminaries

#### Basic biological knowledge

Drug targets refer to the binding sites of drugs in vivo, including proteins enzymes, nucleic acids and others. Drugs act on one or several targets to inhibit the onset of diseases [15]. Drug pathways refer to a series of enzyme catalyzed reaction pathways that transmit molecular signals from outside the cell membrane to exert effects inside the cell. Different targets may exist on the same pathways, and the same target may exist on different pathways. When predicting synergistic drug combinations, targets and pathways are generally important basis.

From known synergistic drug combinations, it can be observed that drugs sharing the same Anatomical Therapeutic Chemical (ATC) classification, exhibiting significant chemical similarity, having comparable indications and side effects, and acting on highly correlated pathways and targets are more likely to form synergistic combinations [16–19].

#### Basic graph theory

Graph is a mathematical structures used to model pairwise relations between objects [20]. A graph is an ordered pair $G = (V, E)$, where $V$ is the set of vertices; $E \subseteq \{xy : x \in V, y \in V\}$ is the

set of edges. For an edge $xy \in E$, the vertex $x, y$ is said to be the end of the edge, and the edge is said to join the two vertices.

The shortest path problem is a classical problem in graph theory, which aims to find the shortest path between any two vertices in a graph. We denote the length of shortest path between vertices $x$ and $y$ in the graph $G$ as $SPL(x, y, G)$. Nowadays, numerous algorithms are available for identifying the shortest path, such as Dijkstra algorithm [21], Bellman-Ford algorithm [22] and Floyd-Warshall algorithm [23]. In this paper, we will use Floyd-Warshall algorithm to find the shortest path due to its simplicity.

The principle of Floyd-Warshall algorithm is dynamic programming [24]. Let $D_{i,j,a_k}$ be the length of the shortest path from $i$ to $j$ with only nodes in the collection $a_k = \{1, 2, ..k\}$ as intermediate nodes:

1. If the shortest path goes through the vertex $k$: $D_{i,j,a_k} = D_{i,k,a_{k-1}} + D_{k,j,a_{k-1}}$;

2. If the shortest path does not go through the vertex $k$: $D_{i,j,a_k} = D_{i,j,a_{k-1}}$;

In a word, $D_{i,j,a_k} = \min(D_{i,j,a_{k-1}}, D_{i,k,a_{k-1}} + D_{k,j,a_{k-1}})$.

The pseudo-code of the Floyd-Warshall algorithm is shown in Algorithm 1, G represents the graph and $|V|$ represents the number of vertices.

**Algorithm 1** Floyd–Warshall Algorithm

```
1: procedure FLOYD(Graph G)
2:    D ← a |V| × |V| array of minimum distances initialized to ∞
      (infinity)
3:    for each edge(u, v) do
4:       D[i][j] ← w(u, v) (the weight of the edge (u, v))
5:    end for
6:    for each vertex v do
7:       D[i][i] ← 0
8:    end for
9:    for k ← 1 to |V| do
10:      for i ← 1 to |V| do
11:        for j ← 1 to |V| do
12:           D[i][j] ← min(D[i][j], D[i][k] + D[k][j])
13:        end for
14:      end for
15:    end for
16:    return D
17: end procedure
```

## Method

In this paper, we reconstructed a drug dataset for angina pectoris and propose a novel algorithm HPRNA to predict synergistic drug combinations for angina pectoris.

### Reconstruction of the drug dataset for angina pectoris

**Some basic datasets and databases.** DDFA(Drug Dataset For Angina): This dataset includes 182 drugs related to the treatment of angina pectoris, identified through expert experience and literature review. It only contains information on these medications and does not include any personal information about patients.

DCDB: DCDB is the first database dedicated to multi-component drug development [25]. The current version of DCDB contains 448555 approved or investing combinatorial drugs involving 2887 single drugs. The drug combinations in the DCDB were collected from PubMed and FDA Orange Book with high confidence. In addition to the corresponding drug

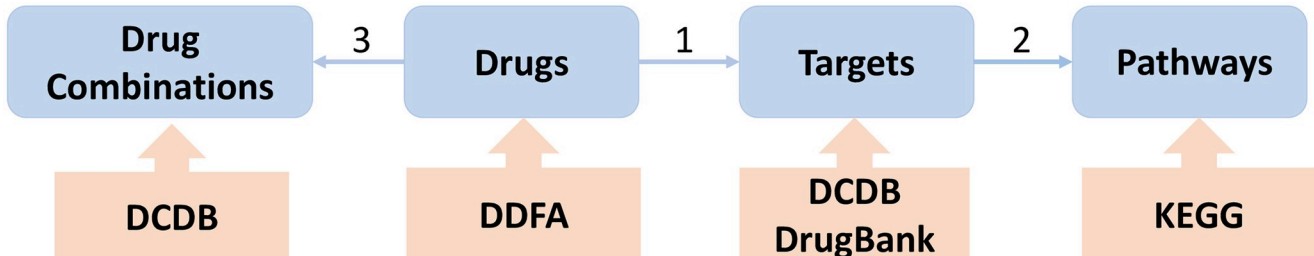

**Fig 1. Reconstruction steps.** The blue section represents the reconstructed data information, while the orange section indicates the database or dataset from which this information originated. Steps 1–3 represent the process of data collection.

combinations information, the DCDB also provides the retrieval function of drug targets. We obtain synergistic drug combinations and targets involved in DDFA from this database.

DrugBank: DrugBank is a comprehensive database that offers detailed information on drugs and drug targets. It provides data on drug properties, chemical structures and mechanisms of action. We get drug related target information from this database.

KEGG(Kyoto Encyclopedia of Genes and Genomes): KEGG is a comprehensive database that integrates information on biological systems, including metabolic and signaling pathways, gene functions, drugs, and diseases. It provides detailed maps and data to facilitate the understanding of complex biological processes, support drug discovery, and analyze disease mechanisms. We download all pathway information related to human from this database to obtain their KEGG ID and the KEGG ID of the corresponding genes on the pathway.

**Reconstruction steps.** Because the information about the target and pathway of the drug is needed in our algorithm, the drugs with missing relevant information are not considered in the algorithm. By eliminating drugs for missing pathways or targets, we used 94 single drugs from DDFA. The steps of reconstructing the dataset are as follows and the simple illustration of the data is shown in Fig 1:

- First, we obtain the target information of the drugs. This information was obtained from the DCDB and DrugBank. We searched the selected drugs for the corresponding targets in the two databases respectively, and considered the union of the two databases' results as the targets corresponding to the drug.

- Second, we obtain pathway information based on the target information. For each drug, we find the pathways corresponding to its targets from KEGG and use this set of pathways as the pathway collection for this drug.

- Finally, we identify synergistic drug combinations involving 94 individual drugs from the DCDB. We retrieve a total of 110 synergistic combinations, of which 12 were ineffective and 4 were unclear.

At this point, we have comprehensive information on heart related drugs. During the process of collecting data, we also looked up their ATC information and chemical expressions from DrugBank, but it was not applied in this algorithm. After the integration, a relatively complete dataset is obtained, which we refer as angina-related Drug Combination Dataset(ADCD).

### Human pathway relationship network algorithm

Chen et al. [14] achieved notable success in synergistic drug prediction by assuming that drugs targeting closely related pathways are more likely to form synergistic pairs. Wang et al. [13] employed a network biology methodology, demonstrating that drugs modulating functionally

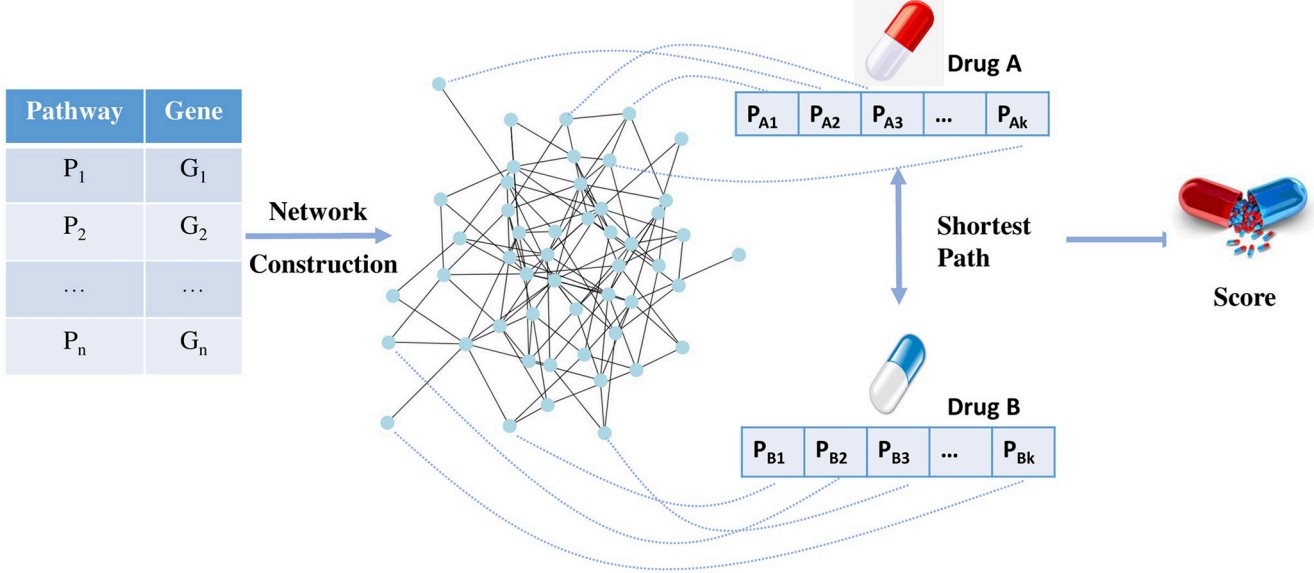

**Fig 2. Human pathway relationship network algorithm flowchart.**

related pathways tend to interact synergistically. Jia et al. [1] investigated established synergistic drug combinations and concluded that drugs acting on related pathways are more inclined to exhibit synergistic effects. Additionally, based on the reconstruction process of our dataset, drug combinations targeting the same pathways reflect interactions with similar targets. Consequently, we assume that drugs acting on closely related or identical pathways are more likely to form effective drug combination. Building on this assumption, HPRNA constructs networks and develops a novel algorithm to facilitate the prediction of drug combinations. The flowchart of HPRNA is shown in Fig 2.

**Build the human pathway relationship network.** The human pathway contains some genetic information. Gene function significantly impacts the activity of pathways and genes with similar functions are often involved in analogous biological pathways. Therefore, gene similarity frequently reflects pathway similarity. When two pathways include a high number of genes with comparable functions, it is probable that these pathways exhibit similar biological roles or mechanisms [26]. Consequently, we construct a network of pathway relationships based on the genes.

Let $P = \{P_1, P_2, P_3, \ldots, P_n\}$ be the collection of human pathways and $G_i = \{g_{i1}, g_{i2}, g_{i3}, \ldots, g_{im_i}\}$ be the collection of genes on human pathway $P_i$. Then, we utilize the gene count within the pathway to conduct Fisher's exact test in order to ascertain any potential relationship between the two pathways. For $P_i$ and $P_j$, let $a_{ij} = |G_i \cap G_j|$, $b_{ij} = |G_i - G_j|$, $c_{ij} = |G_j - G_i|$ and $d_{ij} = M - |G_i \cup G_j|$, Where $M = |G_1 \cup G_2 \cup \cdots \cup G_n|$.

According to the above information, we use Fisher's exact test to calculate the P-value between $P_i$ and $P_j$:

$$p_{ij} = \frac{\binom{a_{ij} + b_{ij}}{a_{ij}}\binom{c_{ij} + d_{ij}}{c_{ij}}}{\binom{a_{ij} + b_{ij} + c_{ij} + d_{ij}}{a_{ij} + c_{ij}}} = \frac{(a_{ij} + b_{ij})!(c_{ij} + d_{ij})!(a_{ij} + c_{ij})!(b_{ij} + d_{ij})!}{a_{ij}!b_{ij}!c_{ij}!d_{ij}!(a_{ij} + b_{ij} + c_{ij} + d_{ij})!} \tag{1}$$

We believe that the two pathways whose P-value is less than $p$ are correlated, and the two pathways whose P-value is greater than $p$ are not correlated, where $p$ is a hyperparameter, $p_{ij}$ is the P-value of $P_i$ and $P_j$. Let $G = (V, E)$ be the human pathway relationship network, the vertex set and edge set of $G$ are represented as follows:

$$V = \{P_i | P_i \in P\}$$

and

$$E = \{(P_i, P_j) | p_{ij} < p\}.$$

**Predictive indicator.** We utilized drug pair scores as a predictive indicator for identifying synergistic drug combinations. A higher score suggests a greater likelihood of synergistic interaction between the drugs. According to the basic assumption, drugs are more likely to form effective combinations when their action pathways are closely related. Therefore, the shorter the distance between the pathways of two drugs, the higher their score should be.

Here, we use the negative exponential function of e to provide specific scores. We defined $A$ and $B$ as two distinct drugs. $P_A = \{P_{A_1}, P_{A_2}, P_{A_3} \ldots P_{A_k}\}$ and $P_B = \{P_{B_1}, P_{B_2}, P_{B_3} \ldots P_{B_k}\}$ refer to the set of pathways associated with drug $A$ and drug $B$. $|P_A|$ and $|P_B|$ refer to the size of set $P_A$ and $P_B$, respectively. If $P_{Ai} \in P_A$ and $P_{Bj} \in P_B$ is accessible in the human pathway relationship network $G$, denoted as $linked(P_{Ai}, P_{Bj}) = 1$. $d(P_{Ai}, P_{Bj})$ refers to the distance between $P_{Ai}$ and $P_{Bj}$. When two drugs act on different pathways, the distance between pathways $P_{Ai}$ and $P_{Bj}$ is defined as the length of the shortest path of the two pathways on the human pathway relationship network $G$, denoted as $SPL(P_{Ai}, P_{Bj}, G)$; when the two drugs act on the same pathway, the distance between them is defined as $C$. Here $C$ is a hyperparameter. In the HPRNA, the score between the two drugs is calculated using the following formula:

$$\mathrm{SCORE(A, B)} = e^{-\dfrac{\displaystyle\sum_{P_{Ai} \in P_A, P_{Bj} \in P_B, linked(P_{Ai}, P_{Bj})=1} d(P_{Ai}, P_{Bj})}{|P_A||P_B|}} \tag{2}$$

$$d(P_{Ai}, P_{Bj}) = \begin{cases} SPL(P_{Ai}, P_{Bj}, G) & if\ P_{Ai} \neq P_{Bj} \\ C & if\ P_{Ai} = P_{Bj} \end{cases} \tag{3}$$

The pseudo-code of prediction score calculation is shown in Algorithm 2, G represents the human pathway relationship network and $P$ represents pathway list, which stores the pathway information of each single drug.

**Algorithm 2** Prediction score

```
1: procedure SCORE(HPR network G, Pathway list P, Drug A and Drug B)
2:    D ← Floyd(G)
3:    x ← 0
4:    for k = 1 to len(P[A - 1]) do
5:      i ← P[A - 1][k - 1]
6:      for l = 1 to len(P[B - 1]) do
7:        j ← P[B - 1][l - 1]
8:        if D_ij == ∞(infinity) then
9:          x ← x
10:       else if D_ij == 0 then
11:         x ← x + C
12:       else
13:         x ← x + D_ij
```

```
14:      end if
15:    end for
16:  end for
17:  x ← x / (len(P[A - 1]) * len(P[B - 1]))
18:  Score(A, B) ← exp(-x)
19:  return Score(A, B)
20: end procedure
```

In HPRNA, varying C values reflect changes in our algorithmic weight, with smaller values signifying higher weights. In ADCD, pathway information is derived from target mappings, where closely related targets typically map to the same pathway. Consequently, identical pathways essentially correspond to identical or closely related targets. Thus, a decrease in the C value effectively increases the weight assigned to targets within our algorithm.

## Experimental results and analysis

### Dataset

Based on the ADCD, the 94 synergistic drug combinations were used as the test positive set, the 12 ineffective drug combinations recorded in DCDB and the 458 different drug combinations after random pairwise combination of 94 single drugs were used as the negative set (the number ratio of positive set to negative set is 1:5).

### Evaluation metric

This paper adopts the receiver operating characteristic curve (ROC) to evaluate the quality of the algorithm. The ROC curve is drawn based on the true positive rate (TPR) and the false positive rate (FPR).

$$TPR = TP/(TP + FN) \tag{4}$$

$$FPR = FP/(FP + TN) \tag{5}$$

Here TP(TN) represents the number of positive(negative) samples are correctly predicted, FN(FP) represents the number of positive(negative) samples that are not correctly predicted. TPR represents the probability of being judged in all positive samples, and FPR represents the probability of being misjudged as positive in all negative samples. The abscissa of the ROC curve is FPR and the ordinate is TPR.

During the process of specific calculation, the score of each sample is taken as the threshold value, the sample with the score greater than the threshold is regarded as positive, and the score less than the threshold is regarded as negative. Each threshold can calculate the corresponding TPR value and FPR value, that is the coordinate point on the ROC. After traversing the score we get the ROC curve. The Area Under the Receiver Operating Characteristic Curve (AUC) is a metric that represents the probability that a randomly chosen positive sample will have a higher score than a randomly chosen negative sample [27]. A higher AUC value indicates better algorithm performance.

### Hyperparameter setting

**Different weights.** The human pathway relationship network was constructed based on gene information from pathways obtained from the KEGG database. In this experiment, we consider two pathways with a Fisher's exact test derived P-value less than 0.001 as correlated, assigning a distance of 1 between them. The effects and analyses of different P-values will be presented below. By statistical calculation, our human pathway relationship network

**Table 1. Comparison of different weights.**

| C | 10 | 5 | 0 | -5 | -10 | -50 | -100 | -150 |
|---|----|---|---|----|-----|-----|------|------|
| **AUC** | 0.53 | 0.57 | 0.62 | 0.64 | 0.65 | 0.68 | 0.68 | 0.68 |

C is a dimensionless scalar, and AUC represents the area under the ROC curve. A higher AUC value indicates better performance of the algorithm.

**Table 2. Comparison of different P-values.**

| P-value | 0.01 | 0.001 | 0.0001 | 0.00001 |
|---------|------|-------|--------|---------|
| AUC | 0.68 | 0.68 | 0.68 | 0.67 |

P-value represents the p-values of Fisher's exact test between genes and AUC represents the area under the ROC curve. A higher AUC value indicates better performance of the algorithm.

encompasses 330 pathways, defining 10765 pairs of related pathways. We do the experiment by changing the hyperparameter *C*, and the experimental results are shown in Table 1.

The performance of the algorithm improves as the C decreases, ultimately converging to a stable value. Through the analysis of the Table 1, the results are basically stable at -100. Therefore, we chose -100 as the definition for the distance between identical pathways.

As can be seen from the Table 1, when we take the basic assumption of WWI [14]: the drugs are more likely to form a synergistic drug combination when they are in a closely linked but not identical pathway, which means the score should be reduced when applied to the same pathway, that is, the defined distance should be increased. This hypothesis is far less effective on our dataset than ours hypothesis: when applied to the same pathway, it is more likely to result in a drug combination.

The discrepancy arises from how we gathered information on drug targets and pathways. Instead of using pathways documented in KEGG for each drug, we used all pathways associated with drug targets as our dataset. Our dataset inherently links pathway information with target information. Drugs that can form synergistic combinations often target closely related targets, which usually act on the same pathways. Thus, assuming that drugs targeting the same pathway hinder combination formation leads to results contrary to expectations. In essence, when using our constructed database as the drug pathway dataset, HPRNA significantly outperforms the results based on the WWI assumption.

**Different P-values.** The results of different human pathway relationship network structures are given in Table 2. We can observed that without significantly reducing the P-value, the distances between pathways in the human pathway relationship network do not undergo substantial changes. As shown in Fig 3, the red and orange curves overlap, indicating similar performance at these p-value thresholds. For the other P-values, the curves nearly coincide, suggesting that the accuracy of the algorithm's predictions does not show significant differences across these thresholds.

This result can be further explained by the findings in the study by Chen et al. [28], which suggest that synergistic drug combinations are frequently associated with specific key biological pathways. These pathways play crucial roles in mediating the effects of drug interactions, and their interactions are often central to the mechanism of synergy. In the context of our algorithm, as long as the relative distance between these key pathways remains relatively stable throughout the computational process, the resulting predictions of drug combinations will not significantly deviate from one another. This is because the underlying biological relationships

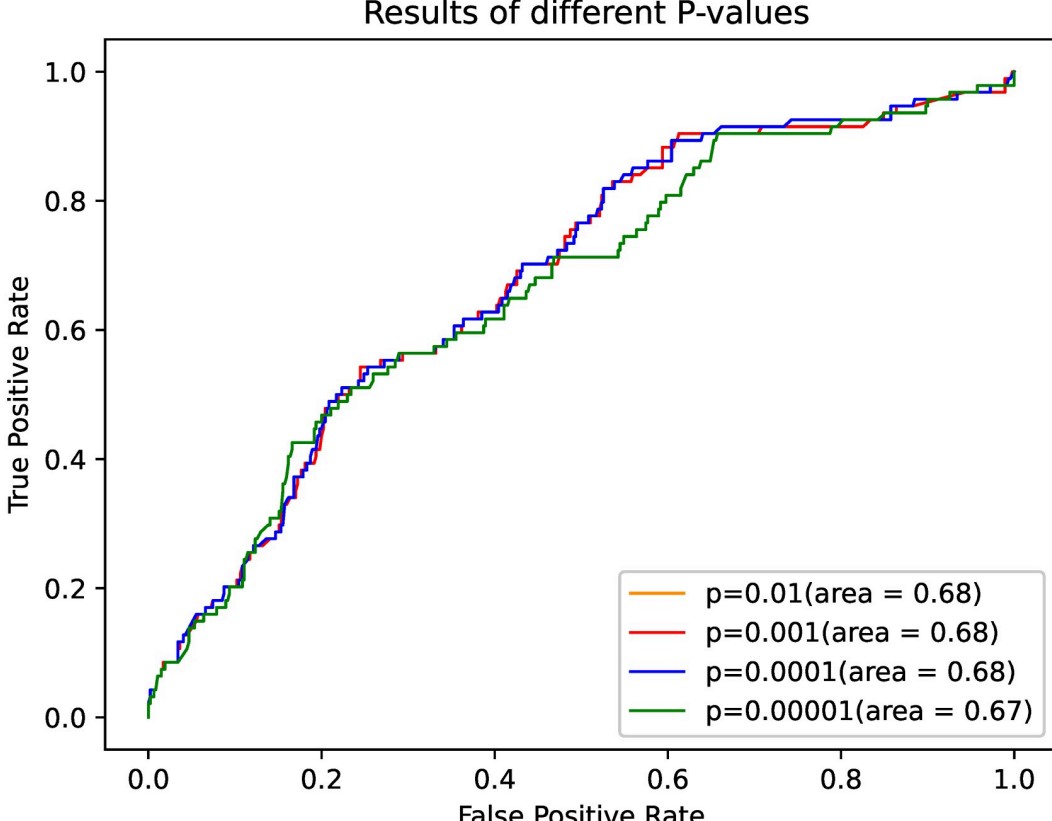

**Fig 3. ROC curves of different P-values.** The x-axis represents the FPR, and the y-axis represents the TPR. Each curve corresponds to a different p-value, with colors indicating the specific p-value threshold.

between the pathways are preserved, ensuring that the synergy predicted by our model aligns closely with the established understanding of these molecular interactions.

## Performance evaluation

We compare and evaluate our methods from two perspectives: data and algorithm. The specific results are detailed below.

**Comparison of different inputs.**  We also consider the comparison of results based on models using input data from two different databases. One model directly considers the pathways documented in KEGG as the drug's pathways, while the other uses the method of utilizing the union of target corresponding pathways as the drug's pathways. The results of the effectiveness of these two input models under different C are provided in Table 3.

**Table 3. Comparison of different inputs.**

| C | 5 | 0 | -5 | -10 | -50 | -100 |
|---|---|---|---|---|---|---|
| AUC of the KEGG input | 0.43 | 0.58 | 0.60 | 0.60 | 0.60 | 0.60 |
| AUC of the HPRNA input | 0.57 | 0.62 | 0.64 | 0.65 | 0.68 | 0.68 |

KEGG input refers to pathway information directly obtained from the KEGG database. HPRNA input refers to pathway information obtained from our reconstructed ADCD.

From the results in the Table 3, it is evident that the results of our dataset are better than those using KEGG data directly.

**Comparison of different algorithms.** In this paper, the HPRNA is used to predict synergistic drug combinations for angina, and its results are compared with the following algorithms:

WWI [14]: This algorithm assumes that synergistic drug combinations are more likely when two drugs target strongly correlated but non-identical pathways. In the WWI algorithm, the score is calculated using the pathway-pathway interaction network for different pathways, and the protein-protein interaction network for the same pathway.

TarOverlap [29]: This algorithm calculates the target association by cosine similarity of two target sets, it is based on the hypothesis that drugs target on the same protein may generate synergistic effect.

TarDis [13]: This algorithm calculates the target association based on the shortest path length between targets on protein-protein interaction network.

PathOverlap: This algorithm calculates the pathway association by cosine similarity of two pathway sets.

PathDis: This algorithm calculates the pathway association based on the shortest path length between pathways on the human pathway relationship network.

The results of the different algorithms are shown in Fig 4. It can be observed that the other algorithms perform worse than HPRNA. This is because the WWI algorithm only utilizes drug-target information when acting on the same pathway and other algorithms rely on a single type of information, either from the target or the pathway alone. In contrast, HPRNA integrates both drug and pathway data during the input process, where drug pathway determination is based on the mapping of drug targets. This approach inherently incorporates information about the correlations between targets, giving HPRNA a significant advantage.

## Validation of HPRNA

We show the top ten scoring drug combinations from HPRNA in Table 4. For the top ten drug combinations based on scores, we searched the literature for experimental information and found that only one prediction was incorrect, and two combinations had not been validated, resulting in a 70% accuracy in effective predictions. This result illustrates the effectiveness of HPRNA.

## Conclusion and future prospects

### Conclusion

The key findings of this study are summarized as follows:

1. Dataset Construction: We successfully integrated multiple diverse datasets to construct a comprehensive resource specifically focused on angina drug treatments. This dataset encompasses essential drug-related information, including target data, pathway details, and chemical structures. It provides a robust foundation for future studies on synergistic drug combinations in the treatment of angina.

2. Development of Human Pathway Relationship Network Algorithm: We developed a novel algorithm that constructs a human pathway relationship network based on genetic similarities between pathways. This approach introduces an innovative method for calculating

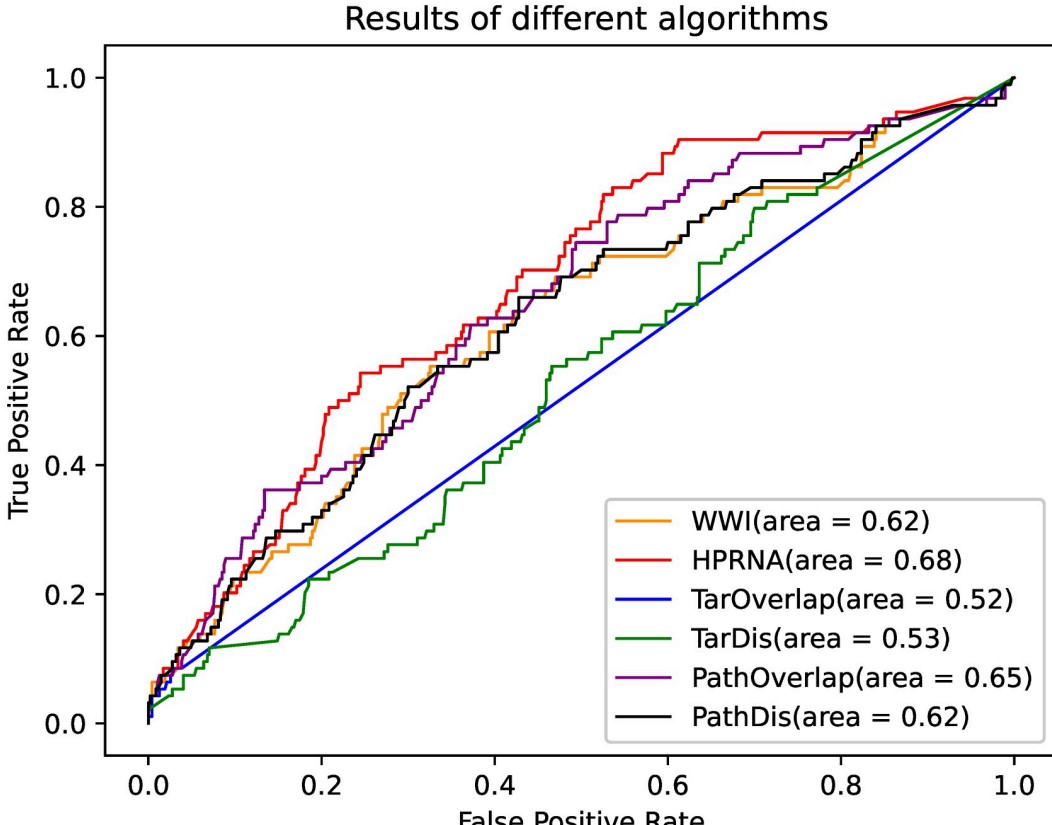

**Fig 4. ROC curves of different algorithms.** The x-axis represents the FPR, and the y-axis represents the TPR. Each curve corresponds to a different method.

drug synergy scores, marking the first effort to predict synergistic drug combinations for angina using pathway-related data. Our findings highlight the critical importance of pathway interactions in identifying effective drug combinations, paving the way for future advancements in this field.

**Table 4. Top 10 score drug combinations.**

| Medicine A | Medicine B | Whether it is a synergistic combination drug | The basis of judgement |
|---|---|---|---|
| Aliskiren | Ramipril | Yes | DCDB |
| Morphine | Naltrexone | Yes | DCDB |
| Propoxyphene | Prednisolone | Yes | Drugbank |
| Atorvastatin | Rosuvastatin | Yes | DCDB |
| Phenylephrine | Pyrilamine | Yes | DCDB |
| Pentazocine | Fimasartan | Unknown | |
| Pyrilamine | Fimasartan | Unknown | |
| Bupivacaine | Fimasartan | No | Drugbank |
| Aliskiren | Valsartan | Yes | DCDB |
| Aliskiren | Losartan | Yes | DCDB |

The basis of judgement refers to the database from which the drug combination information was obtained.

3. Prediction and Validation of Synergistic Drug Combinations: The HPRNA algorithm successfully predicted several promising synergistic drug combinations for angina therapy, demonstrating its reliability and potential for clinical application. Some of the predicted combinations have not yet been experimentally validated, but they offer new perspectives for future research in angina treatment, potentially leading to novel therapeutic strategies that were previously unexplored.

## Future prospects

Research on synergistic drug combinations represents a highly promising area with significant therapeutic potential. The following are key directions for future research that could build upon the findings of this study:

1. Incorporation of Additional Drug Information: To further improve the predictive accuracy of the HPRNA algorithm, future studies could incorporate additional drug-related data, such as side effect profiles, pharmacokinetic properties, and structural similarities. These factors could help refine the algorithm and provide more nuanced predictions for drug combinations.

2. Expanding the Scope of the Algorithm: Currently, our algorithm focuses on predicting synergistic combinations of two drugs. However, many clinical scenarios involve multi-drug regimens. Extending the algorithm to handle combinations of three or more drugs could open up new avenues for more comprehensive therapeutic strategies, providing a deeper understanding of polypharmacy and its impact on angina treatment.

3. Exploration of Machine Learning and Deep Learning Approaches: While the current algorithm is based on deterministic models, incorporating machine learning or deep learning techniques could further enhance its predictive power. These methods are particularly adept at identifying complex, non-linear relationships in large datasets, which could lead to even more accurate and reliable predictions of drug synergy.

4. Clinical Validation and Experimentation: Further work is needed to experimentally validate the predicted drug combinations in clinical settings. Collaborations with pharmacological and clinical researchers will be crucial to assess the real-world efficacy of these predicted combinations, moving the findings from theoretical to practical applications in angina therapy.

5. Personalized Medicine: In the future, it would be valuable to extend the current approach to include individual patient data, such as genetic profiles or specific biomarkers. This would allow for the development of personalized synergistic drug combinations tailored to each patient's unique molecular signature, offering the potential for more targeted and effective treatments.

## Supporting information

**S1 Data. S1 Data consists of three Excel files, each containing different aspects of the drug information related to the treatment of angina: DDFA File contains data on 94 single drugs used for the treatment of angina.** 110 Combination Drugs File includes 110 synergistic drug combinations, each formed by pairing the 94 single drugs from the DDFA file. These combinations have been validated for their therapeutic efficacy in treating angina. The Target

and the Pathway of Target Action File provides detailed information about the molecular targets and associated pathways of action for each of the 94 single drugs in the DDFA file. (ZIP)

## Author Contributions

**Conceptualization:** Guiying Yan.

**Data curation:** Xiangling Zhang, Xiaochun Xing, Yang Li.

**Validation:** Mengyao Zhou, Mengfan Xu, Xiangling Zhang, Xiaochun Xing, Yang Li.

**Visualization:** Mengyao Zhou, Mengfan Xu.

**Writing – original draft:** Mengyao Zhou.

**Writing – review & editing:** Mengyao Zhou, Mengfan Xu, Guanghui Wang, Guiying Yan.

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
