## [Decision Letter · Decision Letter 0]

8 Nov 2024

PONE-D-24-40906HPRNA: Predicting Synergistic Drug Combinations for Angina Pectoris based on Human Pathway Relationship Network AlgorithmPLOS ONE

Dear Dr. Yan,

Thank you for submitting your manuscript to PLOS ONE. After careful consideration, we feel that it has merit but does not fully meet PLOS ONE’s publication criteria as it currently stands. Therefore, we invite you to submit a revised version of the manuscript that addresses the points raised during the review process.

We look forward to receiving your revised manuscript.

Kind regards,

Dr. Vinod

Academic Editor

PLOS ONE

Journal Requirements:

2. Please note that PLOS ONE has specific guidelines on code sharing for submissions in which author-generated code underpins the findings in the manuscript. In these cases, we expect all author-generated code to be made available without restrictions upon publication of the work. 

Please review our guidelines at https://journals.plos.org/plosone/s/materials-and-software-sharing#loc-sharing-code and ensure that your code is shared in a way that follows best practice and facilitates reproducibility and reuse.

6. Please ensure that you refer to Figure 1 in your text as, if accepted, production will need this reference to link the reader to the figure.

Reviewers' comments:

Reviewer's Responses to Questions

**Comments to the Author**

1. Is the manuscript technically sound, and do the data support the conclusions?

Reviewer #1: Yes

Reviewer #2: Partly

Reviewer #3: No

Reviewer #4: No

Reviewer #5: Yes

Reviewer #6: Yes

Reviewer #7: Yes

2. Has the statistical analysis been performed appropriately and rigorously? 

Reviewer #1: Yes

Reviewer #2: Yes

Reviewer #3: I Don't Know

Reviewer #4: No

Reviewer #5: N/A

Reviewer #6: Yes

Reviewer #7: Yes

3. Have the authors made all data underlying the findings in their manuscript fully available?

Reviewer #1: Yes

Reviewer #2: No

Reviewer #3: Yes

Reviewer #4: No

Reviewer #5: Yes

Reviewer #6: Yes

Reviewer #7: Yes

4. Is the manuscript presented in an intelligible fashion and written in standard English?

Reviewer #1: Yes

Reviewer #2: Yes

Reviewer #3: No

Reviewer #4: No

Reviewer #5: Yes

Reviewer #6: Yes

Reviewer #7: No

5. Review Comments to the Author

Reviewer #1: This paper presents a mathematical algorithm designed to predict synergistic drug combinations for the treatment of angina pectoris. The authors have assembled a relevant dataset and introduced a novel indicator. However, I have a few questions regarding their research:

1. Shouldn't the literature review encompass more contemporary studies, particularly those employing deep learning methodologies?

2. How does their approach stack up against machine learning or deep learning-based techniques?

3. It would be beneficial if they could provide a detail of the score for real drug combinations.

Reviewer #2: The author introduce a mathematical algorithm, the Human Pathway Relationship Network Algorithm (HPRNA), which is designed to predict synergistic drug ombinations for angina pectoris. They construct a angina pectoris drug dataset, then construct a comprehensive human pathway network based on the genetic similarity of the pathways which contain information about the targets. However, there are still some questions that need to be improved upon. They should use more data to verify the reliability of the method. In order to clearly present its work, the author should add a detailed method flow chart to facilitate the reader's understanding of the process. In the description of the method, there are some obvious expression errors, which should be carefully checked by the author.

Reviewer #3: I have read the manuscript titled "HPRNA: Predicting Synergistic Drug Combinations for Angina Pectoris based on Human Pathway Relationship Network Algorithm" by Mengyao Zhou et al. with great interest assuming that the manuscript align with my pharmaceutical research domain. But I found the manuscript totally out of my domain and majorily a theoretical paper. As I am not professional in this field, I urge the editor to involve some reviewers from mathematical model research background to do justice.

However, I suggest the authors to work on some of the minor comments along with other expert reviewer's comments:

1. Correct the spelling "caculate"

2. Incomplete sentence "Especially for complex diseases, such as cancer and cardiovascular and cerebrovascular diseases."

3. Cite the text "Compared to single ..... outcomes and fewer adverse effects"

4. Correct the sentence "There are many types of heart disease such as arrhythmia, angina pectoris." Many types??

5. Simplify the sentence "Another extensive study [1] of the molecular mechanisms of drug association suggests that pathway analysis can be an effective study approach for a more comprehensive assessment of drug combination effects."

Reviewer #4: Major issues:

Most of the references are outdated. This shows that the study carried out in the present paper is rely on old information from outdated literature. Thus, authors are strongly advised to rewrite the whole manuscript by considering and comparing the present model with those established in the recent literatures.

In addition, the language of the manuscript is very poor. I have observed several errors, but these are not limited. See below:

1. Abstract: Check spelling of 'caculate', 'investig', and 'judgment'

2. Rephrase "Especially for complex diseases, such as cancer and cardiovascular and cerebrovascular diseases."

3. Incomplete sentence "There are many types of heart 20

disease such as arrhythmia, angina pectoris."

4. which approach? 'However, this approach is costly, time-consuming and poor efficiency.'

5. Mention clearly which parameters are important "Due to the time and economic cost saving, more and more people have turned ....combinations."

6. What is DREAM stands for?

7. Authors must discuss some lines on how the medols are best? 'The two best performing methods in the DREAM challenge of the drug combinations in 2014 are Genomic Residual Effect (DIGRE) and IUPUI CCBB'

8. Poor language 'Due to the time and economic cost saving, more and more people have turned their attention to these methods.' consider rephrasing.

9. Authors should give a comparitive remark of each model such as DIGRE, IUPUL, NLSS etc mentioned in the introduction. And how authors model is better.

10. What is WWI and PPI?

11. Better to use the word 'need' in place of 'urgency'.

12. which sources were used to collect the data and how? ".... integrating data from multiple sources..."

13. Delete the paragraph "The structures of this paper .... future research as it is not required.

14. Cite suitable reference for the paragraph 'Drug targets refer to the binding sites .... generally important basis.'.

15. write a key highlight of the algorithms discussed 'Dijkstra algorithm [17], Bellman-Ford algorithm [18] and Floyd-Warshall algorithm [19].'

16. Useless information 'DrugBank is a comprehensive database that .... related target information from this database.' Authors should focus on how these databases were used to procure the data in present study'

17. Expand KEGG

18. Figure 1 is not cited in the text. Also, the content is unclear. What the sequence 1-3 signify. More details are required in the discussion.

19. why the value -100 was chosen even if the stable results were achieved at -50 also. Is C a unitless quantity?

20. Table 1, 2 and 3 can be combined.

21. Rewrite the discussion for better clarity in more detail "This result can be explained by the results found in the paper of Chen et al. ..... algorithm, the results will not be much different."

22. Rewrite the caption of fig. 2 by inserting more information about the figures.

23. Similarly rewrite the caption of Fig 3 by incorporating significant discussion of the figure.

24. Also include the comparison of literature models in the section 'Comparison of different inputs'. Also discuss how present model is better than those available in literature.

25. A conclusion section must be added in the manuscript that highlight the key discussion of the model described in the present paper. Also, the future prospects are too general. This information must be rewritten and provided within the conclusion section.

26. Authors should rewrite the manuscript by incorporating the most recent model established between 2020-2024 and submit it in a more specialised Mathematics journals.

My recommendation: Reject

Reviewer #5: Synergistic drug combinations therapies have attracted widespread attention due to its advantages of overcoming drug resistance, increasing treatment efficacy and decreasing toxicity. In this paper, the authors introduce a novel mathematical algorithm, the Human Pathway Relationship Network Algorithm (HPRNA), which is designed to predict synergistic drug combinations for angina pectoris. However, I have the following concerns.

1. In the introduction section, the authors missed lots of the latest synergistic combination drugs, and the those mentioned works were developed in many years ago.

2. The model comparison is not convincing, for the authors only adopt one method for comparison. More baselines should be used to make comparison.

3. The conclusion is absent in the paper.

Reviewer #6: This manuscript introduces a novel algorithm, HPRNA, for predicting synergistic drug combinations for angina pectoris, presenting a significant contribution to computational drug design. The paper is well-written, structured effectively, and demonstrates potential in advancing drug combination therapy strategies.

1.The introduction could benefit from a more detailed background on the challenges faced by current drug combination therapies for angina pectoris to better highlight the necessity and novelty of the HPRNA algorithm.

2.Including a separate section or a figure to visually represent the workflow or architecture of the HPRNA would aid in making the algorithmic contributions clearer, especially for readers not familiar with computational models.

3.Updating and expanding the references to include recent studies that have utilized similar computational approaches in drug synergy prediction would better frame the manuscript within the current scientific landscape.

4.A few typos and grammatical errors need correction. For instance, "caculate" in the abstract should be corrected to "calculate."

5.The manuscript indicates that data are available from Drunbank, DCDB, and KEGG. Providing more specific details or direct links to these resources would enhance reproducibility.

Reviewer #7: 1. Compare the proposed algorithm with recently developed algorithms.

2. Choose one or more quality metric for evaluation of proposed algorithms.

3. Rewrite the conclusion section with scientific information.

4. Objectives of the manuscripts require more specific with proposed algorithms.

6. PLOS authors have the option to publish the peer review history of their article (what does this mean?). If published, this will include your full peer review and any attached files.

Reviewer #1: No

Reviewer #2: No

Reviewer #3: No

Reviewer #4: No

Reviewer #5: No

Reviewer #6: No

Reviewer #7: No

---

## [Author Response · Author response to Decision Letter 0]

21 Dec 2024

Response to Reviewer #1

Comment 1: Shouldn't the literature review encompass more contemporary studies, particularly those employing deep learning methodologies? 

Response: Thank you for your suggestion. We have incorporated additional references to deep learning-based approaches in the revised manuscript. However, since our method is a mathematical model based on computational techniques, we have primarily focused on reviewing relevant work related to mathematical and computational models.

Comment 2: How does their approach stack up against machine learning or deep learning-based techniques? 

Response: Thank you for this insightful question. Our approach differs significantly from machine learning and deep learning techniques, particularly when it comes to dealing with sparse labeled data. Machine learning and deep learning methods generally rely on large, labeled datasets for training and learning patterns from the data. However, our dataset is sparse, and the availability of labeled samples is limited, which can hinder the performance of these methods. In contrast, our mathematical model does not require a training phase and it directly calculates the synergy scores for drug combinations using the known drug-target and pathway relationships. This makes our approach more effective and efficient when dealing with sparse data. Additionally, our model requires fewer computational resources compared to deep learning methods, which can be resource-intensive.

Comment 3: It would be beneficial if they could provide a detail of the score for real drug combinations. 

Response: Thank you for your suggestion. We believe that the score is primarily a performance evaluation metric, and its numerical value may not hold significant practical value in isolation. As such, we did not include the detailed scores in the main text. However, we have provided the scores for the top ten drug combinations in the dedicated file for responding to the reviewers, in the table below, for reference.

Response to Reviewer #2

Comment 1: They should use more data to verify the reliability of the method.

Response: Thank you for your valuable comment. Due to limitations in available data, we are currently unable to apply our method to additional datasets. Our approach relies heavily on drug-target and pathway information, which is often not available in many publicly accessible datasets. We hope this clarifies the situation, and we will continue exploring ways to expand the applicability of our approach as more comprehensive datasets become available.

Comment 2: In order to clearly present its work, the author should add a detailed method flow chart to facilitate the reader's understanding of the process.

Response: Thank you for the suggestion. To improve the clarity of the manuscript and help readers better understand the methodology, we have added a detailed flowchart of the method in the revised manuscript. We believe this visual representation will facilitate the comprehension of our approach and provide a clearer overview of the process.

Comment 3: In the description of the method, there are some obvious expression errors, which should be carefully checked by the author.

Response: Thank you for pointing this out. We have carefully reviewed the manuscript and made corrections to the language and expression errors in the method description. The revised manuscript now reflects these improvements, and we have ensured that the wording is clearer and more accurate.

Response to Reviewer #3

Thank you for your review and valuable feedback. We have made the following revisions based on your suggestions:

 Corrected the spelling error and fixed the incomplete sentence.

 Added the appropriate citations and clarified the language where needed.

We appreciate your time and effort in reviewing our manuscript, and your comments have helped improve its quality. Please let us know if you have any further questions or suggestions.

Response to Reviewer #4

Thank you very much for your detailed feedback and valuable suggestions. We have carefully addressed each of your points in the revised manuscript. Below is a point-by-point response to your comments:

Comment 1: Abstract: Check spelling of 'caculate', 'investig', and 'judgment'

Response: We appreciate the reviewer’s comments on the language. We have carefully reviewed the manuscript and made the necessary corrections, including fixing spelling and grammatical errors. Specific changes include:

 "caculate" corrected to "calculate"

 "investig" corrected to "investigate"

 "judgment" corrected to "judgement".

Additionally, we have revised several sentences for clarity and readability throughout the manuscript.

Comment 2: Rephrase: "Especially for complex diseases, such as cancer and cardiovascular and cerebrovascular diseases."

Response: We have rephrased this sentence as follows: "New drug research and development faces complex processes, high economic costs, and challenges such as transient resistance, especially for complex diseases like cancer, cardiovascular diseases, and cerebrovascular disorders." This revision maintains the original meaning while improving the clarity and flow of the sentence.

Comment 3: Incomplete sentence: "There are many types of heart disease such as arrhythmia, angina pectoris."

Response: We have corrected this sentence to: "There are several types of heart disease, including arrhythmia and angina pectoris." This revision ensures that the sentence is complete and clear.

Comment 4: Which approach? "However, this approach is costly, time-consuming and poor efficiency."

Response: We have rephrased this sentence as follows: "However, these traditional experimental methods are costly, time-consuming, and have low efficiency."

Comment 5: Mention clearly which parameters are important: "Due to the time and economic cost saving, more and more people have turned ....combinations."

Response: We have revised this sentence to clarify the role of mathematical models in drug screening: "With the rapid advancement of computational technologies, various mathematical model-based methods for screening drug combinations have emerged. These computational approaches are becoming increasingly popular due to their ability to save time and reduce costs."

Comment 6: What does DREAM stand for?

Response: DREAM Challenges (dreamchallenges.org) are collaborative competitions that pose important biomedical questions to the scientific community and evaluate participants’ predictions in a statistically rigorous and unbiased way, emphasizing model reproducibility and methodological transparency.

Comment 7: Authors must discuss some lines on how the models are best: 'The two best performing methods in the DREAM challenge of the drug combinations in 2014 are Genomic Residual Effect (DIGRE) and IUPUI CCBB.'

Response: We have expanded the discussion on the DREAM challenge and the best-performing models: "In the 2014 DREAM challenge on drug combinations, the DIGRE and IUPUI_CCBB models were among the best-performing methods, both relying on deterministic mathematical formulations. DIGRE effectively utilized a residual effect approach to model the complex relationships between genomic data and drug responses, providing accurate predictions based on mathematical principles. Similarly, the IUPUI_CCBB model employed a robust mathematical framework that integrated multiple biological layers, optimizing drug efficacy predictions through sophisticated mathematical modeling."

Comment 8: Poor language: 'Due to the time and economic cost saving, more and more people have turned their attention to these methods.' Consider rephrasing.

Response: We have rephrased this sentence as follows: "With the rapid advancement of computational technologies, various mathematical model-based methods for screening drug combinations have emerged. These computational approaches are becoming increasingly popular due to their ability to save time and reduce costs."

Comment 9: Authors should give a comparative remark on each model, such as DIGRE, IUPUI, NLSS, etc. mentioned in the introduction. And how the authors' model is better.

Response: We appreciate this suggestion and fully acknowledge the importance of comparing our model with existing models such as DIGRE, IUPUI, and NLSS. However, due to the limited data available on angina pectoris in our study, as well as the differences in the datasets used by these models, a direct comparison was not feasible at this stage. Nonetheless, we plan to conduct a comparative analysis with these models once additional data is collected in the future. We will include such comparisons in a subsequent version of the study to further validate the performance of our model.

Comment 10: What is WWI and PPI?

Response: We have clarified the terms "WWI" and "PPI" as follows: "WWI refers to a method for predicting drug combinations, which is discussed in the Performance Evaluation section. PPI refers to Protein-Protein Interaction networks, which are crucial for understanding the biological interactions between proteins."

Comment 11: Better to use the word 'need' in place of 'urgency'.

Response: We have replaced the term "urgency" with "need" as suggested, to better reflect the context of the discussion.

Comment 12: Which sources were used to collect the data and how? ".... integrating data from multiple sources..."

Response: In the "Reconstruction of the drug dataset for angina pectoris" section, we have provided more detailed information about the data sources and how the data were integrated into the study.

Comment 13: Delete the paragraph "The structures of this paper.... future research as it is not required."

Response: We have deleted the paragraph "The structures of this paper..." as it was not necessary and did not contribute to the manuscript.

Comment 14: Cite suitable references for the paragraph 'Drug targets refer to the binding sites... generally important basis.'

Response: We have added relevant citations to support the statement regarding drug targets and their binding sites.

Comment 15: Write a key highlight of the algorithms discussed (Dijkstra algorithm [17], Bellman-Ford algorithm [18], and Floyd-Warshall algorithm [19]).

Response: Here are the key highlights of the algorithms discussed:

1. Dijkstra Algorithm: Efficient for finding the shortest path from a single source to all other nodes in a graph with non-negative edge weights. 

2. Bellman-Ford Algorithm: Can handle graphs with negative weight edges and detect negative weight cycles. 

3. Floyd-Warshall Algorithm: A dynamic programming-based algorithm for finding the shortest paths between all pairs of nodes in a graph. It works for both positive and negative edge weights and has a time complexity of O(V^3), making it suitable for dense graphs or small to medium-sized networks.

Comment 16: Useless information: 'DrugBank is a comprehensive database that... related target information from this database.' Authors should focus on how these databases were used to procure the data in the present study.

Response: We have removed unnecessary details about DrugBank and focused on how it was used in this study to obtain drug-target information relevant to our model.

Comment 17: Expand KEGG.

Response: Thank you for your comment. KEGG stands for Kyoto Encyclopedia of Genes and Genomes. We have already included its full name in the article.

Comment 18: Figure 1 is not cited in the text. Also, the content is unclear. What do the sequences 1-3 signify? More details are required in the discussion.

Response: We have correctly cited Figure 1 in the text and clarified that the sequences 1-3 represent the steps involved in data collection. Additional explanations have been added to the discussion to provide more details about the figure.

Comment 19: Why was the value -100 chosen even if stable results were achieved at -50? Is C a unitless quantity?

Response: C is the hyperparameter of the model. The effect of choosing -50 or -100 for this parameter is essentially the same, and we randomly selected -100 for convenience. In fact, selecting -50 would not have impacted the results of our model. Additionally, we have stated that 'C' is a unitless quantity, as it is a hyperparameter in our model and does not correspond to any physical units.

Comment 20: Table 1, 2, and 3 can be combined.

Response: We believe that Tables 1 and 2, which focus on the impact of different hyperparameters on the algorithm, should remain separate from Table 3, which assesses the effects of different input data on the model’s performance. We believe keeping them separate provides clearer presentation.

Comment 21: Rewrite the discussion for better clarity: "This result can be explained by the results found in the paper of Chen et al... algorithm, the results will not be much different."

Response: We have rewritten the discussion for clarity: "This result can be further explained by the findings in the study by Chen et al., which suggest that synergistic drug combinations are often associated with specific key biological pathways. These pathways are crucial in mediating the effects of drug interactions and are often central to the mechanism of synergy. In the context of our algorithm, as long as the relative distance between these key pathways remains stable throughout the computational process, the resulting predictions of drug combinations will not significantly deviate from one another. This is because the underlying biological relationships between the pathways are preserved, ensuring that the synergy predicted by our model aligns closely with the established understanding of these molecular interactions."

Comment 22: Rewrite the caption of Fig. 2 by inserting more information about the figures.

Response: We have revised the caption for Fig. 2 to provide more detailed information about the figure’s content and its relevance to the manuscript.

Comment 23: Similarly, rewrite the caption of Fig. 3 by incorporating significant discussion of the figure.

Response: We have updated the caption for Fig. 3 to include a more detailed explanation of the figure, providing context.

Comment 24: Also, include the comparison of literature models in the section 'Comparison of different inputs'. Also, discuss how the present model is better than those available in the literature.

Response: We have added several comparative models in Comparison of different algorithms. In this part, the control model remains unchanged while the input data is varied to demonstrate the accuracy and effectiveness of the data we have collected.

Comment 25: A conclusion section must be added in the manuscript that highlights the key discussion of the model described in the present paper. Also, the future prospects are too general. This information must be rewritten and provided within the conclusion section.

Response: We have added a conclusion section summarizing the key findings of our study. This section also provides more specific directions for future research, moving beyond general statements to focus on concrete future goals.

Comment 26: Authors should rewrite the manuscript by incorporating the most recent models established between 2020-2024 and submit it to a more specialized Mathematics journal.

Response: We appreciate the reviewer’s comment. In our recent literature search, we found that many of the recent models for drug combination prediction (from 2020 to 2024) are based on deep learning and machine learning approaches. These methods generally require large, labeled datasets for supervised learning. However, the dataset available for our study is relatively small and does not contain sufficient labeled information to apply these techniques effectively. Moreover, given the limited scope of the available data, we have opted to use a mathematical modeling approach.

Although we have not directly compared our model with these newer machine learning or deep learning methods, we acknowledge their importance and potential. We have added relevant references to recent studies on machine learning and deep learning methods in the "Related Work" section to provide context and highlight the distinction between our approach and

---

## [Decision Letter · Decision Letter 1]

15 Jan 2025

HPRNA: Predicting Synergistic Drug Combinations for Angina Pectoris based on Human Pathway Relationship Network Algorithm

PONE-D-24-40906R1

Dear Dr. Yan,

We’re pleased to inform you that your manuscript has been judged scientifically suitable for publication and will be formally accepted for publication once it meets all outstanding technical requirements.

Kind regards,

Vinod Kumar Vashistha

Academic Editor

PLOS ONE

Additional Editor Comments (optional):

Reviewers' comments:

Reviewer's Responses to Questions

**Comments to the Author**

1. If the authors have adequately addressed your comments raised in a previous round of review and you feel that this manuscript is now acceptable for publication, you may indicate that here to bypass the “Comments to the Author” section, enter your conflict of interest statement in the “Confidential to Editor” section, and submit your "Accept" recommendation.

Reviewer #1: All comments have been addressed

Reviewer #2: All comments have been addressed

Reviewer #5: All comments have been addressed

Reviewer #6: All comments have been addressed

Reviewer #7: All comments have been addressed

2. Is the manuscript technically sound, and do the data support the conclusions?

Reviewer #1: Yes

Reviewer #2: Yes

Reviewer #5: Yes

Reviewer #6: Yes

Reviewer #7: Yes

3. Has the statistical analysis been performed appropriately and rigorously? 

Reviewer #1: Yes

Reviewer #2: Yes

Reviewer #5: N/A

Reviewer #6: Yes

Reviewer #7: Yes

4. Have the authors made all data underlying the findings in their manuscript fully available?

Reviewer #1: Yes

Reviewer #2: Yes

Reviewer #5: Yes

Reviewer #6: Yes

Reviewer #7: Yes

5. Is the manuscript presented in an intelligible fashion and written in standard English?

Reviewer #1: Yes

Reviewer #2: Yes

Reviewer #5: Yes

Reviewer #6: Yes

Reviewer #7: Yes

6. Review Comments to the Author

Reviewer #1: Thank you for your response, I have no further questions.

Reviewer #2: The author has responded to the relevant questions, most of the questions were well answered. There are no more questions.

Reviewer #5: (No Response)

Reviewer #6: All issues that I concerned have been addressed. I thank the authors for their effort to revise the manuscript considering my comments.

Reviewer #7: In the present state, authors have addressed the answers of all questions and the revised version of the manuscript consider for publication of the Journal.

7. PLOS authors have the option to publish the peer review history of their article (what does this mean?). If published, this will include your full peer review and any attached files.

Reviewer #1: No

Reviewer #2: No

Reviewer #5: No

Reviewer #6: **Yes: **Yu-An Huang

Reviewer #7: No

---

## [Editor Report · Acceptance letter]

28 Jan 2025

PONE-D-24-40906R1 

PLOS ONE

Dear Dr. Yan, 

I'm pleased to inform you that your manuscript has been deemed suitable for publication in PLOS ONE. Congratulations! Your manuscript is now being handed over to our production team.

Kind regards, 

on behalf of

Dr. Vinod Kumar Vashistha 

Academic Editor

PLOS ONE